# Learning Hyperbolic Representations for Unsupervised 3D Segmentation

## Abstract

There exists a need for unsupervised 3D segmentation on complex volumetric data, particularly when annotation ability is limited or discovery of new categories is desired. Using the observation that much of 3D volumetric data is innately hierarchical, we propose learning effective representations of 3D patches for unsupervised segmentation through a variational autoencoder (VAE) with a hyperbolic latent space and a proposed gyroplane convolutional layer, which better models the underlying hierarchical structure within a 3D image. We also introduce a hierarchical triplet loss and multi-scale patch sampling scheme to embed relationships across varying levels of granularity. We demonstrate the effectiveness of our hyperbolic representations for unsupervised 3D segmentation on a hierarchical toy dataset, BraTS whole tumor dataset, and cryogenic electron microscopy data.

## 1 Introduction

Recent advances in technology have greatly increased both the availability of 3D data, as well as the need to process and learn from 3D data. In particular, technologies such as magnetic resonance imaging and cryogenic electron microscopy (cryo-EM) have led to greater availability of 3D voxel data. Deep learning is a promising technique to do so, but producing annotations for 3D data can be extremely expensive, especially for richer tasks such as segmentation in dense voxel grids. In some cases, labels may also be impossible to produce due to the limitations of current knowledge, or may introduce bias if we want to conduct scientific discovery. Unsupervised learning, which does not require annotations, is a promising approach for overcoming these limitations.

In this work, we tackle the challenging problem of unsupervised segmentation on complex 3D voxel data by addressing the essential challenge of representation learning. We expand from prior literature in the hyperbolic domain that conducts classification in simple data to the task of segmentation in 3D images, which requires significantly more representation discriminability. In order to learn effective representations, we need to capture the structure of our input data. We observe that 3D images often have inherent hierarchical structure: as a biomedical example, a cryo-EM tomogram of a cell has a hierarchy that at the highest level comprises the entire cell; at a finer level comprises organelles such as the mitochondria and nucleus; and at an even finer level comprises sub-structures such as the nucleolus of a nucleus or proteins within organelles. For downstream analysis, we are typically interested in the unsupervised discovery and segmentation of structures spanning multiple levels of hierarchy. However, prior work on representation learning for unsupervised 3D segmentation does not explicitly model hierarchical structure between different regions of a 3D image. We argue that this hampers the ability to leverage hierarchical relationships to improve segmentation in complex 3D images.

Our key insight is that we can utilize a hyperbolic embedding space to learn effective hierarchical representations of voxel regions in 3D images. Hyperbolic representations have been proposed as a continuous way to represent hierarchical data, as trees can be embedded in hyperbolic space with arbitrarily low error (Sarkar, 2011). These methods have shown promise for modeling data types such as natural language word taxonomies (Nickel & Kiela, 2017; 2018), graphs (Nickel & Kiela, 2017; Mathieu et al., 2019; Ovinnikov, 2019; Chami et al., 2019), as well as simple MNIST (LeCun et al., 2010) image data for classification (Mathieu et al., 2019). To the best of our knowledge, our work is the first to introduce learning hyperbolic representations to capture hierarchical structure

among subregions of complex 3D images, and to utilize the learned hyperbolic representations to perform a complex computer vision task such as segmentation.

Our approach for learning hyperbolic representations of 3D voxel grid data is based on several key innovations. First, to handle larger and more complex 3D data such as biomedical images, we propose a hyperbolic 3D convolutional VAE along with a new gyroplane convolutional layer that respects hyperbolic geometry. Second, we enhance our VAE training objective with a novel self-supervised hierarchical triplet loss that helps our model learn hierarchical structure within the VAE's hyperbolic latent space. Finally, since our goal in segmentation is to learn hierarchy within voxel regions of 3D input, we present a multi-scale sampling scheme such that our 3D VAE can simultaneously embed hierarchical relationships across varying levels of granularity.

In summary, our key contributions are as follows:

- We introduce a hyperbolic 3D convolutional VAE with a novel gyroplane convolutional layer that scales the learning of hyperbolic representations to complex 3D data.

- We propose a multi-scale sampling scheme and hierarchical triplet loss in order to encode hierarchical structure in the latent space and perform 3D unsupervised segmentation.

- We demonstrate the effectiveness of our approach through experiments on a synthetic 3D toy dataset, the Brain Tumor Segmentation (BraTS) dataset (Menze et al., 2014; Bakas et al., 2017; 2018), and cryo-EM data.

## 2    RELATED WORK

**Segmentation on 3D voxel data**    Since 3D voxel grids are dense, computer vision tasks such as supervised segmentation are commonly performed using deep learning architectures with 3D convolutional layers (Chen et al., 2016; Dou et al., 2017; Hesamian et al., 2019; Zheng et al., 2019). However, due to the challenges of obtaining voxel-level segmentations in 3D, there has been significant effort in finding semi-supervised approaches, including using labels only from several fully annotated 2D slices of an input volume (Çiçek et al., 2016), using a smaller set of segmentations with joint segmentation and registration (Xu & Niethammer, 2019), and using one segmented input in conjunction with other unlabelled data (Zhao et al., 2019).

Unsupervised approaches for 3D segmentation are useful not only for further reducing the manual annotation effort required, but also for scientific discovery tasks where we lack the sufficient knowledge to provide representative training examples for structures of interest. Moriya et al. (2018) extends to 3D data an iterative approach of feature learning followed by clustering (Yang et al., 2016). Nalepa et al. (2020) uses a 3D convolutional autoencoder architecture and performs clustering of the latent representations. Another approach, (Dalca et al., 2018), uses a network pre-trained on manual segmentations from a separate dataset to perform unsupervised segmentation of 3D biomedical images. However, this limits applicability to areas where we already have a dataset with manual annotations and makes it unsuitable for unbiased unsupervised discovery. Gur et al. (2019) and Kitrungrotsakul et al. (2019) developed unsupervised methods for 3D segmentation of vessel structures, but these are specialized and do not generalize to the segmentation of other structures. Beyond unsupervised 3D segmentation, there has been work such as Ji et al. (2019) that performs unsupervised 2D segmentation based on a mutual information objective, and Caron et al. (2018), which proposes using the clustered output of an encoder as pseudo-labels. While these methods can be applied to 2D slices of a 3D volume to perform 3D segmentation, they generally suffer limitations due to insufficient modeling of the 3D spatial information. None of the aforementioned approaches explicitly model hierarchical structure, which is the main focus of our work.

**Hyperbolic representations**    A recent line of work has employed hyperbolic space to model hierarchical structure, with the intuition that tree structures can be naturally embedded into continuous hyperbolic space (Nickel & Kiela, 2017). Several works have proposed hyperbolic variational autoencoders (VAEs) as an unsupervised method to learn hyperbolic representations. Ovinnikov (2019) proposes a Wasserstein autoencoder on the Poincaré ball model of hyperbolic geometry. Nagano et al. (2019) proposes a VAE on the hyperboloid model of hyperbolic geometry where the last layer of the encoder is an exponential map, and derives a reparametrisable sampling scheme for the wrapped normal distribution, which they use for the prior and posterior. Mathieu et al. (2019)

proposes a VAE on the Poincaré ball model of hyperbolic geometry. In addition to having the last layer of the encoder be an exponential map, Mathieu et al. (2019) also proposes to have the first layer of the decoder be the gyroplane layer proposed by Ganea et al. (2018) in order to better handle the geometry of the hyperbolic latent space, and applies their model to MNIST image classification. Our work differs by introducing an approach for learning hyperbolic representations that models the hierarchy between sub-volumes of complex 3D images, and uses a novel hierarchical triplet loss and sampling scheme to capture relationships among multiple levels of granularity in a given input.

In addition, a related field of study has sought to generalize traditional Euclidean neural networks or their components to non-Euclidean spaces. Ganea et al. (2018) proposes hyperbolic feed-forward and recurrent architectures based on the theory of gyrovector spaces. Building on this work, Chami et al. (2019) propose a hyperbolic graph convolutional network. Other works such as Bachmann et al. (2019); Becigneul & Ganea (2019); Gu et al. (2019) have also proposed learning with a product space of manifolds. Our work generalizes a layer of Ganea et al. (2018) in order to create and use a new hyperbolic convolutional layer, which we call the gyroplane convolutional layer.

## 3 PRELIMINARIES

**Hyperbolic Space**    Hyperbolic space is a non-Euclidean space with constant negative curvature. Curvature is a measure of the deviation of the geometry from a flat plane (Chami et al., 2019). There are five equivalent models of hyperbolic geometry. Following previous work (Mathieu et al., 2019; Ganea et al., 2018; Lou et al., 2020), we use the Poincaré ball model. Hyperbolic space can be considered the continuous version of trees (Nickel & Kiela, 2017), making it a natural choice for embedding hierarchical data. Trees can be embedded in the Poincaré ball with arbitrarily low error (Sarkar, 2011), and like the leaves of a tree, the area of a disc in the Poincaré ball increases exponentially with the radius. Unlike trees, hyperbolic space is smooth, permitting deep learning.

**Poincaré ball model of hyperbolic geometry**    The Poincaré ball (of curvature $c = -1$) is the open ball of radius 1 centered at the origin equipped with the *metric tensor* $\mathfrak{g}_p = (\lambda_x)^2 \mathfrak{g}_e$, where the conformal factor $\lambda_x = \frac{2}{1-||x||^2}$ and $\mathfrak{g}_e$ is Euclidean metric tensor (i.e., the usual dot product). Formally, this makes the Poincaré ball a *Riemannian manifold*. The distance $\mathbf{d_p}$ between points on the Poincaré ball is given by:

$$\mathbf{d_p}(x, y) = \cosh^{-1}\left(1 + 2\frac{||x-y||^2}{(1 - ||x||^2)(1 - ||y||^2)}\right) \tag{1}$$

The exponential and logarithm maps are a useful way to map from Euclidean space to the Poincaré ball and vice versa (in general, to map from a tangent space to a Riemannian manifold and vice versa). On the Poincaré ball, the exponential and logarithm maps have the closed forms

$$\exp_z(v) = z \oplus \left(\tanh\left(\frac{\lambda_z ||v||}{2}\right)\frac{v}{||v||}\right), \log_z(y) = \frac{2}{\lambda_z}\tanh^{-1}(|| - z \oplus y||)\frac{-z \oplus y}{|| - z \oplus y||} \tag{2}$$

where $\oplus$ denotes *Mobius addition*, which was first introduced by Ungar (2001) as a way to define vector operations on hyperbolic space (see Appendix).

## 4 METHODS

In this section, we describe our approach for learning hyperbolic representations of subvolumes (3D patches) from 3D voxel grid data. We propose a model that comprises a 3D convolutional variational autoencoder (VAE) with hyperbolic representation space and a new gyroplane convolutional layer, along with a novel hierarchical triplet loss and a multi-scale sampling scheme that facilitates learning hierarchical structure within the hyperbolic latent space. To produce segmentations, we cluster the learned hyperbolic representations. In Section 4.1, we describe our VAE framework as well as our proposed gyroplane convolutional layer and hierarchical triplet loss. In Section 4.2, we introduce our approach of hyperbolic clustering for segmentation.

### 4.1 Unsupervised hyperbolic representation learning

**3D Hyperbolic VAE framework** The VAE framework (Kingma & Welling, 2013; Rezende et al., 2014) is widely used for unsupervised representation learning, but requires new innovations to lean effective hierarchical representations in 3D image data. Our proposed hyperbolic VAE consists of a 3D convolutional encoder which maps sampled 3D patches of the input volume into hyperbolic space and produces the parameters of the variational posterior, as well as a 3D convolutional decoder which reconstructs the patch from sampled latent hyperbolic representations. The last layer of the encoder is an exponential map that ensures that the output is in hyperbolic space, and the first layer of the decoder is our proposed gyroplane convolutional layer which maps hyperbolic space to Euclidean space. This ensures that both the encoder and decoder respect the hyperbolic geometry of the latent space. We use the wrapped normal distribution as our prior and posterior distribution (see Appendix). Figure 1 illustrates an overview of this VAE framework.

Our variational autoencoder takes as input a patch of fixed size $m \times m \times m$. This allows our model to learn representations of subvolumes that can subsequently be used to perform voxel-level segmentation of the whole 3D volume. To learn hierarchical structure in the 3D scene of each input, we generate training examples using a multi-scale sampling scheme that samples patches of size $r \times r \times r$, where $r$ is randomly sampled. We use two sampling schemes, one for input of smaller sizes and one for input of larger sizes. In both schemes, for a given 3D volume, we sample $i$ patch centers $v_i$ uniformly.

In the sampling scheme for smaller inputs, we sample $r \sim \mathcal{U}(\ell_{\min}, \ell_{\max})$, where $\ell_{\min}, \ell_{\max}$ are hyperparameters. The patch is then upsampled or downsampled to size $m \times m \times m$. For larger inputs, we observe that semantic changes tend to occur on a logarithmic scale, so we instead first sample $e \sim \mathcal{U}(\ell_{\min}, \ell_{\max})$ and then set $r = 2^e$. This sampling scheme is motivated by the intuition that for larger patches, a small change in $r$ is less likely to correspond to significant semantic difference.

**Gyroplane convolutional layer** Since $\mathbb{R}^n = \mathbb{R} \times \ldots \times \mathbb{R}$, high-dimensional Euclidean spaces can be decomposed into a product of low-dimensional Euclidean spaces. An equivalent decomposition does not hold for arbitrary Riemannian manifolds, making it difficult to generalize the usual (Euclidean) convolutional layer to arbitrary Riemannian manifolds. For manifolds that are products of manifolds, we can generalize the usual convolution by replacing the Euclidean affine transformation with an affine transformation on the manifold. For the Poincaré ball, one analogue of the Euclidean affine transformation is the gyroplane operator $f_{a,p}$ (see Appendix). The details are as follows: for simplicity, suppose $x$ is a 4D tensor containing elements of the Poincaré ball and our kernel size is $k \times k \times k$, with an odd $k$ value. Our gyroplane convolutional layer is defined as:

$$y_{r,s,t} = \sum_{\alpha=r-\lfloor k/2 \rfloor}^{r+\lfloor k/2 \rfloor} \sum_{\beta=s-\lfloor k/2 \rfloor}^{s+\lfloor k/2 \rfloor} \sum_{\gamma=t-\lfloor k/2 \rfloor}^{t+\lfloor k/2 \rfloor} f_{a,p}(x_{\alpha,\beta,\gamma}) \tag{3}$$

Our gyroplane convolutional layer can be extended in the same way as Euclidean convolutional layers to incorporate even kernel size $k$, input and output channels, padding, stride, and dilation. Our model's encoder mean output ($\mu$ in Figure 1) can be interpreted as a product of Poincaré balls, justifying our definition and use of the gyroplane convolutional layer.

**Hierarchical triplet loss** As our model is trained on patches of the whole 3D volume, the hierarchical structure of the volume is not readily apparent from the individual inputs. To help the model infer hierarchical structure, we provide self-supervision in the form of a hierarchical triplet loss where positive examples are sub-patches of an anchor patch and negative examples are patches that do not overlap with the anchor patch.

To sample 3D patches for the triplet loss, we first generate an anchor patch centered at voxel $v$ with size $r \times r \times r$ according to one of the above sampling schemes. A positive child patch is generated as a smaller sub-patch of the anchor patch as follows: the positive child patch is centered at $v$ with size $r_{\text{child}} \times r_{\text{child}} \times r_{\text{child}}$, where $r_{\text{child}} \sim \mathcal{U}(\ell_{\min}, r - r_{\text{gap}})$, and $r_{\text{gap}}$ is a hyperparameter representing the gap in size between the anchor size and the child size. A negative child is a patch of size $r_{\text{child}} \times r_{\text{child}} \times r_{\text{child}}$ centered at $v_{\text{neg}}$, where $v_{\text{neg}}$ is sampled uniformly from the set of voxels $w$ such that a patch of size $r_{\text{child}} \times r_{\text{child}} \times r_{\text{child}}$ centered at $w$ does not overlap with the anchor patch.

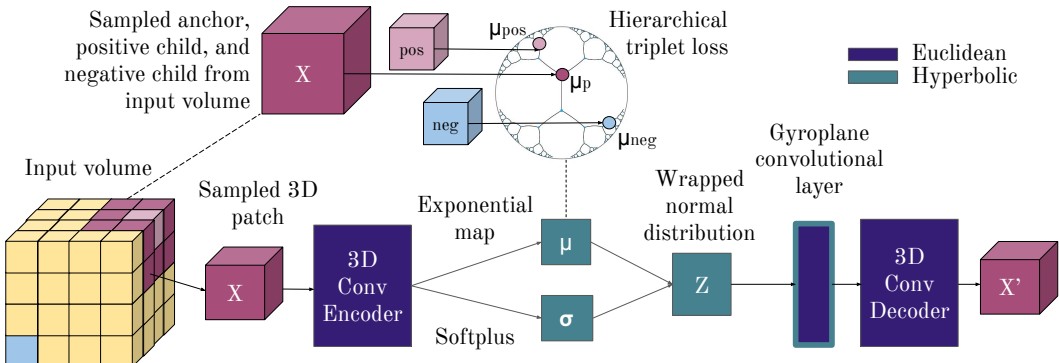

Figure 1: Our method learns hyperbolic representations of subvolumes of 3D voxel grid data, through a 3D hyperbolic VAE with a gyroplane convolutional layer. We enhance the VAE training objective with a self-supervised hierarchical triplet loss that facilitates learning hierarchical structure within the VAE's hyperbolic latent space.

Our choice of positive and negative patches is motivated by the compositional hierarchy of 3D volumes. Our hierarchical triplet loss encourages the anchor patch and a sub-patch (positive child) to have similar representations, while encouraging the anchor patch and a distant patch (negative child) to have dissimilar representations. In hyperbolic space, this has the interpretation of belonging to the same hierarchy and belonging to different hierarchies, respectively. We learn hierarchy within a 3D image through this triplet loss.

The hierarchical triplet loss can be formulated with any dissimilarity measure $\mathbf{d}$ between the encoder outputs $\mu$ (see Figure 1) of the anchor $\mu_{\mathrm{p}}$, positive child $\mu_{\mathrm{pos}}$, and negative child $\mu_{\mathrm{neg}}$. For our model, we take $\mathbf{d}$ to be the Poincaré ball distance $\mathbf{d_p}$ and define our triplet loss with margin $\alpha$ as:

$$L_{\mathrm{triplet}}(\mu_{\mathrm{p}}, \mu_{\mathrm{pos}}, \mu_{\mathrm{neg}}) := \max(0, \mathbf{d_p}(\mu_{\mathrm{p}}, \mu_{\mathrm{pos}}) - \mathbf{d_p}(\mu_{\mathrm{p}}, \mu_{\mathrm{neg}}) + \alpha) \tag{4}$$

This formulation can be extended to any metric space by taking the dissimilarity measure $\mathbf{d}$ to be the space's metric. In particular, for our ablations using an Euclidean latent space we take the dissimilarity measure $\mathbf{d}$ to be the Euclidean distance.

**Optimization**   We optimize a loss function that can be decomposed as an evidence lower bound (ELBO) loss and our new hierarchical triplet loss that encourages the learning of hierarchical structure in the latent representations. The total loss can be formulated as $L_{\mathrm{total}} = L_{\mathrm{ELBO}} + \beta L_{\mathrm{triplet}}$, where $\beta$ is a hyperparameter that controls the strength of the triplet loss.

### 4.2 SEGMENTATION BY CLUSTERING REPRESENTATIONS

**Hyperbolic clustering**   In 3D segmentation, we seek to assign each voxel $v$ a segmentation label $s_{\mathrm{v}} \in \{1, \ldots, n\}$, where $n$ is the number of segmentation classes. We perform segmentation by clustering the representations of patches centered at each voxel. We first generate latent representations $\mu_{\mathrm{v}}$ for each voxel $v$ by running our trained VAE on a patch of fixed size $p \times p \times p$ centered at $v$, upsampled or downsampled to encoder input size $m \times m \times m$ if necessary. We then cluster the $\mu_v$ into $n$ clusters, and produce a segmentation by assigning each $v$ the cluster label of $\mu_v$. Clustering is done using a $k$-means algorithm that respects hyperbolic geometry, which we derive by replacing the Euclidean centroid and distance computations of classical $k$-means with their appropriate counterparts in Riemannian geometry, the Fréchet mean and manifold distance. We calculate the Fréchet mean using the algorithm of Lou et al. (2020).

## 5   EXPERIMENTS

Though our method is general to any 3D voxelized grid data, we evaluate on several biomedical datasets due to the availability of annotated 3D voxel data in the field. We evaluate our method quantitatively on both a synthetic 3D toy dataset simulating biological image data, as well as the

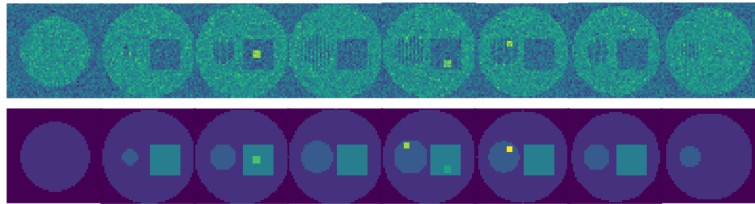

Figure 2: Sampled 2D slices from a 3D volume in our biologically-inspired toy dataset. The top row showcases the raw input data, and the bottom row showcases the ground truth segmentation.

BraTS tumor segmentation dataset. Our biologically-inspired toy dataset allows quantitative evaluation of segmentation at multiple levels of hierarchy, while the BraTS dataset is a well-known benchmark for 3D MRI segmentation. We also demonstrate the use of unsupervised segmentation for discovering new biological features in real-world cryo-EM data.

For all models, the encoder of our variational autoencoder is comprised of four 3D convolutional layers with kernel size $5$ of increasing filter depth $\{16, 32, 64, 128\}$. The decoder is of the same structure, except with decreasing filter depth and a gyroplane convolutional layer as the initial layer. We use $\beta = 1e3$ as the weighting factor between $L_{\text{ELBO}}$ and $L_{\text{triplet}}$ and $\alpha = 0.2$ as the triplet margin, and train the model using the Adam optimizer (Kingma & Ba, 2014). We fix the representation dimension to be $d = 2$. For training on the toy dataset, we sample 3D volume sizes uniformly, and for BraTS and the cryo-EM dataset we sample using an exponential scale (see Section 4.1). For inference, we obtain the latent representations of $5 \times 5 \times 5$ patches densely across the full volume, and then perform hyperbolic $k$-means clustering, where the number of clusters $k$ is a hyperparameter that controls the granularity of the segmentation. For quantitative evaluation, we then use the Hungarian algorithm (Kuhn, 1955) to match each predicted segmentation class with a corresponding ground truth label.

## 5.1 BIOLOGICALLY-INSPIRED TOY DATASET

Since most 3D image datasets are not annotated at multiple levels of hierarchy, we first generate a hierarchical toy dataset to enable more thorough evaluation of the effectiveness of our model for unsupervised 3D segmentation. We note that datasets such as ShapeNet (Chang et al., 2015) are unsuitable since they contain 3D shape models instead of 3D voxel grid data, which is the focus of our work. Our toy dataset is inspired by cryo-EM tomograms of cells. Each volume in our toy dataset contains multiple levels of hierarchy with objects at each level determined by texture and size. Figure 2 shows an example input volume with sampled slices shown.

Each 3D image of our toy dataset consists of a background and a large sphere which represents a cell, which we will refer to as *Level 1* of the image's hierarchy. The large sphere contains a medium-size sphere and cube meant to represent cellular substructures such as vesicles, which we will refer to as *Level 2*. In turn, each of these shapes contains two smaller objects of the same shape in *Level 3*. The color, size, and location of each shape vary randomly. We also apply biologically realistic noise in the form of pink noise. More details can be found in the Appendix.

To measure the ability of our model to capture the hierarchy of the toy dataset, we separately evaluate on the three levels of hierarchy defined above and use the average class DICE score to compare segmentation performance. Since our model is unsupervised, segmentation classes are assigned to ground truth labels using the Hungarian algorithm. See results in Table 1 and Table 8.

**Comparison with prior approaches** Table 1 shows quantitative comparison of our method with prior state-of-the-art 3D unsupervised, 2D unsupervised (which we extend to 3D), and semi-supervised models. As unsupervised 3D segmentation is a relatively unexplored field, we provide these baselines with different levels of supervision for additional reference. Çiçek et al. (2016) was trained with $2\%$ of the ground truth slices in each of the $xy$, $yz$, and $xz$ planes, and Zhao et al. (2019) was trained with one fully annotated atlas. Ji et al. (2019) was implemented using the authors' original code and extrapolated to 3D. For Nalepa et al. (2020) and Moriya et al. (2018), we re-implemented their methods as the original code was unavailable. Our model performs signifi-

Table 1: Comparison with prior approaches on toy dataset

|  | Dice *Level 1* | Dice *Level 2* | Dice *Level 3* | Supervision type |
|---|---|---|---|---|
| Çiçek et al. (2016) | 0.96843 | 0.82867 | 0.66771 | 3D Semi-supervised |
| Zhao et al. (2019) | 0.98892 | 0.65478 | 0.35651 | 3D Semi-supervised |
| Nalepa et al. (2020) | 0.53017 | 0.27612 | 0.11997 | 3D Unsupervised |
| Ji et al. (2019) | 0.58866 | 0.29086 | 0.14999 | 2D to 3D Unsupervised |
| Moriya et al. (2018) | 0.62777 | 0.31120 | 0.14131 | 3D Unsupervised |
| **Ours** | **0.95211** | **0.54065** | **0.21623** | 3D Unsupervised |

Table 2: Ablation studies on toy dataset

| Latent Space | Configuration | Dice *Level 1* | Dice *Level 2* | Dice *Level 3* |
|---|---|---|---|---|
| Euclidean | Base | 0.78370 | 0.32170 | 0.10890 |
|  | Triplet | 0.76111 | 0.34202 | 0.15349 |
| Hyperbolic | Base | 0.83231 | 0.35185 | 0.13528 |
|  | GyroConv | 0.90468 | 0.47297 | 0.20363 |
|  | Triplet | 0.94522 | 0.53392 | **0.22217** |
|  | GyroConv & Triplet | **0.95211** | **0.54065** | 0.21623 |

cantly better at all levels of hierarchy compared to its unsupervised counterparts, and comparably to the semi-supervised approach of Zhao et al. (2019).

**Ablation**  Table 8 presents ablation studies on the hierarchical toy dataset comparing our contributions: Euclidean vs. hyperbolic representations, the addition of our gyroplane convolutional layer, and the addition of our hierarchical triplet loss. The Base Euclidean configuration is the 3D convolutional VAE with Euclidean latent space, no gyroplane convolutional layer, and trained with just the ELBO loss. The Triplet Euclidean configuration adds the hierarchical triplet loss to the base Euclidean configuration. The Base Hyperbolic configuration is the same as the Base Euclidean configuration except with hyperbolic latent space. The Triplet configuration is the hyperbolic analogue of the Euclidean Triplet configuration, and GyroConv configurations have the addition of the gyroplane convolutional layer.

Hyperbolic representations outperform their Euclidean counterparts in all experiments. We attribute this to the more efficient and better organization of hyperbolic representations. When we introduce the hierarchical triplet loss, performance improves significantly for our hyperbolic models, but performance for our Euclidean model does not improve as much, likely due to information loss in representing hierarchical input. Introducing the gyroplane convolutional layer shows clear improvement over our Base Hyperbolic model, which shows the benefit of having a layer that respects the geometry of the latent space. The combination of the triplet loss and gyroplane convolutional layer exhibits the most gain over the Base Hyperbolic model, but only small gains over the model with just the added triplet loss. This shows the importance of the our triplet loss for learning effective hierarchical representations.

## 5.2 Brain Tumor Segmentation challenge dataset

The BraTS 2019 dataset is a public, well-established benchmark dataset containing 3D MRI scans of brain tumors along with per-voxel ground truth annotations of tumor segmentation masks. The scans are of dimension $200 \times 200 \times 155$ and have four modalities; we use the FLAIR modality, which is the most commonly used one-modality input. We use the same evaluation metric as in the BraTS challenge, and compare DICE score on whole tumor (WT) segmentation, which is detectable solely from FLAIR. There are 259 high grade glioma (HGG) labelled training examples, which we split into 180 train examples, 39 validation examples, and 40 test examples. We do not use the official validation or test sets because the ground truth annotations for these sets are not publicly available. Table 3 shows the comparison of our results against prior work; we train all baselines on the specified data split for fair comparison. The only exception is the current state-of-the-art fully-

supervised result (Jiang et al., 2019) in Table 3, which also uses all 4 modalities. We show this for reference as an upper bound; the reported number is trained on the full train set and evaluated on the BraTS test set.

Our best model performs significantly better than the unsupervised baselines, and in addition outperforms one 3D semi-supervised model. This illustrates the ability of our hyperbolic latent representations to effectively capture the hierarchical structure in individual brain scans. We use a granular segmentation with three clusters for quantitative evaluation in order to capture the tumor, brain, and background, then use the Hungarian algorithm for assignment. In addition, we also show qualitative results for our model (see Figure 3), which include byproduct segmentations from the same model with different numbers of clusters specified, showcasing additionally discovered features in the scan that could also be clinically useful.

Table 3: Table shows comparison on BraTS 2019 dataset. Figure shows a qualitative example where top left image is a slice from a 3D test volume, and the three other images show segmentations with $2, 3, 4$ numbers of clustering centroids respectively, illustrating multiple levels of hierarchy learned.

| BraTS dataset | Dice WT | Algorithm type |
|---|---|---|
| SOTA (Jiang et al., 2019) | 0.88796 | 3D Fully-supervised |
| Zhao et al. (2019) | 0.64826 | 3D Semi-supervised |
| Çiçek et al. (2016) | 0.75965 | 3D Semi-supervised |
| Ji et al. (2019) | 0.21076 | 2D to 3D Unsupervised |
| Moriya et al. (2018) | 0.42515 | 3D Unsupervised |
| Nalepa et al. (2020) | 0.49503 | 3D Unsupervised |
| **Ours** | **0.68391** | 3D Unsupervised |

## 5.3 CRYOGENIC ELECTRON MICROSCOPY TOMOGRAMS

Finally, we show an example of unsupervised 3D segmentation in a real-world scenario where unsupervised discovery is important. Cryogenic electron microscopy is a technique that images cells at cryogenic temperatures with a beam of electrons. The value of each voxel is the electron density at that location, and is created through reconstruction from tilt slices of $\pm 70$ degrees from electron microscopy. Cryo-EM tomograms are a rich source of biological data, capturing many subcellular features that are unknown or unexplored. We train our model on three $512 \times 512 \times 250$ cryo-E0M tomograms of cells collected from a research laboratory, and run inference on a fourth tomogram. Figure 3 shows segmentations produced by our model on a mitochondria from the evaluation tomogram, using the proposed hyperbolic embedding space vs. Euclidean embedding space, and at a coarse and finer level of granularity. Unlike the Euclidean approach, the hyperbolic approach discovers a fine-grained class corresponding to small features on the mitochondria, which may be macromolecular aggregates. As an example of performing unsupervised discovery with our model, the discovered features can now be investigated for their chemical identities and functions.

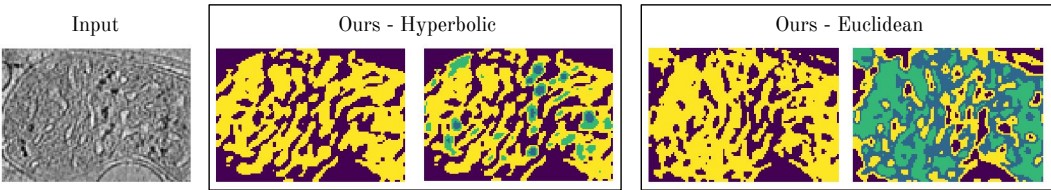

Figure 3: Leftmost image is a partial slice from a 3D cryo-EM image. The middle box shows segmentation from our best hyperbolic model, the rightmost box shows segmentation from our best Euclidean model. The segmentations in each box correspond to clustering using 2 vs. 4 classes.

## 6 CONCLUSION

We propose a method for learning hyperbolic representations of subvolumes in 3D voxel grid data, that is based on a hyperbolic 3D convolutional VAE with a new gyroplane convolutional layer that respects hyperbolic geometry. We enhance the VAE training objective with a self-supervised hierarchical triplet loss that facilitates learning hierarchical structure within the VAE's hyperbolic latent space, and a multi-scale sampling scheme. We demonstrate that hyperbolic clustering of learned voxel-level representations can be used to achieve state-of-the-art unsupervised 3D segmentation, on a hierarchical toy dataset and the BraTS dataset. We also illustrate the promise of using our model for unsupervised scientific discovery on an example of cryogenic electron microscopy data.

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

# A    APPENDIX

## A.1    RIEMANNIAN MANIFOLDS

In this section we give a more complete introduction to Riemannian manifolds, of which hyperbolic space is an example. Riemannian manifolds are spaces that locally resemble Euclidean space. To define this mathematically, we first introduce a *manifold* as a set of points $\mathcal{M}$ that locally resembles the Euclidean space $\mathbb{R}^n$. Associated with each point $\mathbf{x} \in \mathcal{M}$ is a vector space called the *tangent space* at $\mathbf{x}$, denoted $\mathcal{T}_{\mathbf{x}}\mathcal{M}$, which is the space of all directions a curve on the manifold $\mathcal{M}$ can tangentially pass through point $\mathbf{x}$. A metric tensor $\mathfrak{g}$ defines an inner product $\mathfrak{g}_{\mathbf{x}}$ on every tangent space, and a *Riemannian manifold* is a manifold $\mathcal{M}$ together with a metric tensor $\mathfrak{g}$. Distance on a Riemannian manifold as can defined as the following. Let $\gamma : [a, b] \to \mathcal{M}$ be a curve on the manifold $\mathcal{M}$. The *length* of $\gamma$ is defined to be $\int_a^b |\gamma'(t)|_{\gamma(t)} dt$ and denoted $L(\gamma)$. The *distance* between any two points $\mathbf{x}, \mathbf{y}$ on the manifold is defined as $d_{\mathcal{M}}(\mathbf{x}, \mathbf{y}) = \inf L(\gamma)$, where the inf is taken over all curves $\gamma$ that begin at $\mathbf{x}$ and end at $\mathbf{y}$. This distance makes $\mathcal{M}$ a metric space.

The *exponential map* $\exp_{\mathbf{x}}(v) : \mathcal{T}_{\mathbf{x}}\mathcal{M} \to \mathcal{M}$ is a useful way to map vectors from the (Euclidean) tangent space to the manifold. The exponential map is defined as $\exp_{\mathbf{x}}(v) = \gamma(1)$, where $\gamma$ is the unique geodesic, the shortest possible curve between two points, starting at $\mathbf{x}$ with starting direction $v$. Intuitively, one can think of the exponential map as telling us how to travel one step starting from a point $\mathbf{x}$ on the manifold in the $v$ direction. The logarithmic map $\log_v(x) : \mathcal{M} \to \mathcal{T}_{\mathbf{x}}\mathcal{M}$ is the inverse of the exponential map, and maps vectors back to Euclidean space.

## A.2    GYROVECTOR OPERATIONS IN THE POINCARÉ BALL

Gyrovector operations were first introduced by Ungar (2008) to generalize the Euclidean theory of vector spaces to hyperbolic space. Mobius addition is the Poincaré ball analogue of vector addition in Euclidean spaces. The closed-form expression for Mobius addition on the Poincaré ball with negative curvature $c$ is Mathieu et al. (2019):

$$z \oplus_c y = \frac{(1 + 2c\langle z, y \rangle + c||y||^2)z + (1 - c||z||^2)y}{1 + 2c\langle z, y \rangle + c^2||z||^2||y||^2} \tag{5}$$

As one might hope anticipate, when $c = 0$ we recover Euclidean vector addition. Additionally, the analogue of Euclidean vector subtraction is Mobius subtraction, which is defined as $x \ominus_c y = x \oplus_c (-y)$, and the analogue of Euclidean scalar multiplication is Mobius scalar multiplication, which can be defined for a scalar $r$ as (Ganea et al., 2018):

$$r \otimes_c x = \frac{1}{\sqrt{c}} \tanh(r \tanh^{-1}(\sqrt{c}||x||)) \frac{x}{||x||} \tag{6}$$

where we also recover Euclidean scalar multiplication when $c = 0$. In this paper, we only consider the Poincaré ball with fixed constant negative curvature $c = 1$, which allows us to drop the dependence on $c$.

## A.3    WRAPPED NORMAL DISTRIBUTION

The importance of the normal distribution in Euclidean space has led to many attempts to generalize the normal distribution to Riemannian manifolds. The wrapped normal distribution is one popular way to do this (Mathieu et al., 2019; Nagano et al., 2019). The wrapped normal distribution can be defined on an arbitrary Riemannian manifold as the push-forward measure obtained by mapping the normal distribution in Euclidean space along the manifold's exponential map. The probability density function of the wrapped normal with mean $\mu$ and covariance $\Sigma$ is:

$$\mathcal{N}_P(z|\mu, \Sigma) = \mathcal{N}_E(\lambda_\mu(z)|0, \Sigma) \left( \frac{\mathbf{d_P}(\mu, z)}{\sinh(\mathbf{d_P}(\mu, z))} \right) \tag{7}$$

where the subscripts $P, E$ indicate whether the distribution is over the Poincaré ball or Euclidean space, respectively. To use the wrapped normal in a VAE, we require both a way to sample from the wrapped normal as well as a way to train its parameters. Mathieu et al. (2019) provides a reparametrization and sampling scheme for the wrapped normal on the Poincaré ball.

### A.4 GYROPLANE OPERATOR

The gyroplane layer can be thought of as a hyperbolic affine transformation, and is motivated by the fact we can express a Euclidean affine transformation as $\langle a, z-p \rangle = \text{sign}(\langle a, z-p \rangle)||a||\mathbf{d_E}(z, H_{a,p})$ (Ganea et al., 2018), where $\mathbf{d_E}$ is Euclidean distance and $H_{a,p} = \{z \in \mathbb{R}^p | \langle a, z-p \rangle = 0\}$. $H_{a,p}$ is called the decision hyperplane. Ganea et al. (2018) defined the gyroplane operator $f_{a,p}$ from this formulation by replacing each component with its hyperbolic equivalent:

$$f_{a,p}(z) = \text{sign}\left(\langle a, \log_p(z) \rangle_p\right) |a|_p \mathbf{d_p}(z, H_{a,p}) \tag{8}$$

where $H_{a,p}$ is the hyperbolic decision boundary $H_{a,p} = \{z \in \mathcal{B} | \langle a, \log_p(z) \rangle = 0\}$, and the distance to the hyperbolic decision boundary $\mathbf{d_p}(z, H_{a,p})$ is

$$\mathbf{d_p}(z, H_{a,p}) = \sinh^{-1}\left(\frac{2|\langle -p \oplus z, a \rangle|}{(1 - ||-p \oplus z||^2)||a|}\right) \tag{9}$$

### A.5 TOY DATASET

Our biologically-inspired toy dataset has 120 total volumes, which we split into 80 train examples, 20 validation examples, and 20 test examples. Each toy volume in our dataset is $50 \times 50 \times 50$ and contains multiple levels of hierarchy.

The first level of hierarchy (*Level 1*) is a an outer sphere centered in the volume of radius $r \sim \mathcal{N}(25, 1)$. Using a cell analogy, this represents the outer cell. The second level (*Level 2*) consists of spheres ("vesicles") and cuboids ("mitochondria"), both of which are textured, hollow, and completely contained within the outer cell wall. The positions are randomly sampled with radius of $r \sim \mathcal{N}(8, 0.5)$ and with side length of $s \sim 2 \cdot \mathcal{N}(8, 0.5)$. In the third level (*Level 3*) we introduce small spheres and cuboids ("proteins") in the vesicle spheres and mitochondria cubloids respectively. The *Level 3* proteins are randomly positioned with radius of $r \sim \mathcal{N}(2, 0.2)$ and with side length of $s \sim 2 \cdot \mathcal{N}(3, 0.15)$.

Each instance of a shape with a particular size is also given its own unique texture to mimic the different organelles of the cell. The color of each object is chosen randomly, according to a standard normal distribution. We also apply pink noise with magnitude $m = 0.25$ to the volume as it is commonly seen in biological data.

In addition, we have also added a biologically-inspired toy dataset with irregular shapes for evaluating datasets with different characteristics. This dataset was created through applying smooth noise to the boundaries of each shape. Specifically, we generate smooth noise by first sampling random points in our voxel grid and random values according to a Gaussian distribution, and interpolate to get a smooth noise. We then use this smooth noise function to perturb the points that fall within the interior of the three largest shapes. See an example of the dataset in Figure 4.

We demonstrate our method's performance in comparison to prior work on the aforementioned irregular dataset in Table 4, and an ablation study applied on the same irregular dataset in Table 5, both with error bars over four independent runs.

We note that in Table 4, our proposed method outperforms prior work significantly on the irregular dataset, following our initial observations from Table 1 to show state-of-the-art performance. We can see that while all methods show slight decrease in performance, our method is still able to maintain the lead in performance as compared to prior work across all levels.

For ablations on the irregular toy dataset in Table 5, we find that our best models with hyperbolic latent space still outperform models with Euclidean latent space, as with our original toy dataset. We also demonstrate that the gyroplane convolutional layer and hierarchical triplet loss are both effective compared to the base hyperbolic configuration. However, despite it being effective compared to the base hyperbolic configuration, models with hyperbolic hierarchical triplet loss performed less well across the board as compared to the original toy dataset. We hypothesize that this is due to the specific challenges that the irregular dataset brings, for example, needing to recognize noisy instances of irregular shape as the same class in Levels 2 and 3. Therefore, our proposed gyroplane convolutional layer by itself is able to add more effective learning capacity, and shows significant improvement. The added hierarchical triplet loss performs less well on the irregular dataset than in our original toy dataset because in our multi-patch sampling method, each patch is sampled at

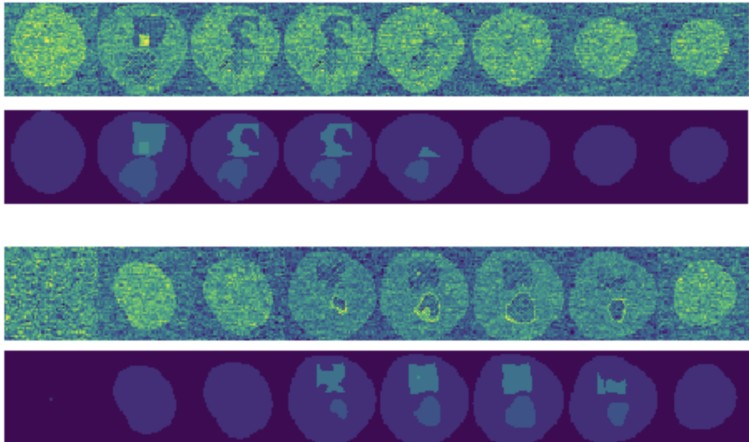

Figure 4: Sampled 2D slices from two examples of 3D volumes in our irregular biologically-inspired toy dataset, showing large variance in shapes across input. For each 3D volume example, the top row showcases the raw input data, and the bottom row showcases the ground truth segmentation.

Table 4: Comparison with prior approaches on irregular toy dataset

|  | Dice *Level 1* | Dice *Level 2* | Dice *Level 3* | Supervision type |
|---|---|---|---|---|
| Çiçek et al. (2016) | 0.93459 | 0.76226 | 0.65480 | 3D Semi-supervised |
| Zhao et al. (2019) | 0.97308 | 0.63275 | 0.30164 | 3D Semi-supervised |
| Nalepa et al. (2020) | 0.51008 | 0.24476 | 0.09324 | 3D Unsupervised |
| Ji et al. (2019) | 0.53370 | 0.24476 | 0.18865 | 2D to 3D Unsupervised |
| Moriya et al. (2018) | 0.54804 | 0.23930 | 0.13181 | 3D Unsupervised |
| **Ours** | **0.950** $\pm0.006$ | **0.474** $\pm0.063$ | **0.202** $\pm0.026$ | 3D Unsupervised |

random capturing parts of the 3D input. Since the boundary of the shape changes in every image, with random sampling learning is more difficult for our hierarchical triplet loss. We don't see the same phenomenon for Level 1 since background/foreground segmentation is an easier task. We conclude that with the level of irregularity added to our dataset (see examples in Figure 4), the gyroplane convolutional layer with the hyperbolic latent space provides more effectiveness than the triplet loss.

We also note that in real-world datasets, such as in our work in cryogenic electron microscopy, the overall shapes of each class of object is similar, and do not contain such dramatic irregularity. For example, vesicles are almost-circular ellipses with only slight eccentricity (deformations with slight stretch), but without distinctive irregularities and protrusions in our irregular dataset. Overall, our experiments demonstrate that different components of our method are useful for different scenarios, and that our method overall robustly outperforms prior work across data with different characteristics. All hyperbolic configurations of our method seen in Table 4 outperform past unsupervised methods, and our approach of learning hyperbolic representations of complex 3D data for segmentation is more effective than methods with canonical Euclidean representations.

Last, for runtime on the toy datasets, our implementations of the proposed models take between five to eight hours to train on a single Titan RTX GPU for both Euclidean and Hyperbolic variants. We note that for our current implementation, hyperbolic k-means clustering takes on the order of a few hours versus minutes for Euclidean k-means. However, this is because we are using our own unoptimized implementation based on recent research in fast Frechet mean algorithms, and standard packages such as scikit-learn do not include hyperbolic k-means algorithms. The Euclidean k-means algorithms in these packages are heavily optimized with parallelization. We anticipate that such optimization would bring the hyperbolic k-means's runtime to the order of the Euclidean k-means, as the computational complexity of the algorithms are similar in practice.

Table 5: Ablation studies on irregular toy dataset

| Latent Space | Configuration | Dice *Level 1* | Dice *Level 2* | Dice *Level 3* |
|---|---|---|---|---|
| Euclidean | Base | $0.705 \pm 0.161$ | $0.349 \pm 0.102$ | $0.148 \pm 0.066$ |
| | Triplet | $0.906 \pm 0.037$ | $0.506 \pm 0.078$ | $0.201 \pm 0.061$ |
| Hyperbolic | Base | $0.845 \pm 0.123$ | $0.413 \pm 0.129$ | $0.180 \pm 0.052$ |
| | GyroConv | $0.947 \pm 0.006$ | $\mathbf{0.530} \pm 0.008$ | $\mathbf{0.220} \pm 0.007$ |
| | Triplet | $\mathbf{0.951} \pm 0.002$ | $0.471 \pm 0.011$ | $0.192 \pm 0.005$ |
| | GyroConv & Triplet | $0.950 \pm 0.006$ | $0.474 \pm 0.063$ | $0.202 \pm 0.026$ |

## A.6 BRATS DATASET

We also conduct an ablation study on the BraTS dataset with each of our added components with error bars over 4 independent runs. Results are shown in Table 6. We can see that our best Hyperbolic model outperforms our best Euclidean model significantly. The addition of the triplet loss improved both Euclidean and Hyperbolic models, while the Hyperbolic models see more improvement due to ability to encode hierarchy in hyperbolic latent space. Our gyroplane convolutional layer also improves performance, while both of our additions jointly improve upon our Hyperbolic baseline, showing the benefit of these added components to learning effective representations.

Table 6: Ablation study for BraTS dataset. We report the mean and standard deviation of DICE scores for 4 independent runs.

| Latent Space | Configuration | Dice |
|---|---|---|
| Euclidean | Base | $0.388 \pm 0.022$ |
| | Triplet | $0.517 \pm 0.050$ |
| Hyperbolic | Base | $0.414 \pm 0.017$ |
| | GyroConv | $0.539 \pm 0.014$ |
| | Triplet | $0.610 \pm 0.028$ |
| | GyroConv & Triplet | $\mathbf{0.692} \pm 0.009$ |

We include the average and 95 percentile Hausdorff distance as complementary evaluation metrics on the BraTS dataset. See Table 7. We show performance on our method compared to other unsupervised baselines; our model outperforms all prior methods on both metrics.

Table 7: Comparison of our method against prior unsupervised work in Hausdorff distance.

| | Average Hausdorff | 95% Hausdorff |
|---|---|---|
| Moriya et al. (2018) | 118.1439 | 170.434 |
| Ji et al. (2019) | 96.865 | 114.400 |
| Nalepa et al. (2020) | 87.704 | 110.803 |
| **Ours** | **77.940** | **97.641** |

## A.7 EVALUATION

We use DICE score to quantitatively evaluate segmentation performance. The DICE score is defined as the following:

$$DICE = \frac{2TP}{2TP + FN + FP} \tag{10}$$

where $TP$ is the number of true positives, $FN$ is the number of false negatives, and $FP$ is the number of false positives. For our toy dataset, we first assign predicted classes to ground truth labels using the Hungarian algorithm Kuhn (1955), then evaluate using the average class DICE score. For the BraTS dataset Menze et al. (2014); Bakas et al. (2017; 2018), we evaluate DICE of the whole tumor segmentation following official evaluation guidelines.

We also use Hausdorff distance to evaluate the worst-case performance of our model. For two sets of points $A$, $B$, the directed Hausdorff distance from $A$ to $B$ is defined as

$$h(A, B) = \max_{a \in A} \left\{ \min_{b \in B} \mathbf{d}(a, b) \right\} \tag{11}$$

where $\mathbf{d}$ is any distance function. We will take $\mathbf{d}$ to be the Euclidean distance. The Hausdorff distance is then defined to be

$$H(A, B) = \max \left\{ h(A, B), h(B, A) \right\} \tag{12}$$

The official BraTS evaluation uses 95 percentile Hausdorff distance as measure of model robustness (Bakas et al., 2018).

### A.8 HYPERPARAMETERS

We use a single set of hyperparameters on all of our evaluation datasets, and these hyperparameters are not tuned on any of the evaluation datasets. In order to obtain a reasonable set of hyperparameters, we created a completely separate synthetic dataset on which we trained models and tuned hyperparameters. This synthetic dataset was created in a similar manner to our toy dataset; however, we designed it to have different and fewer objects, simpler nesting structure, no noise, and fewer textures. The application of this single set of hyperparameters to our evaluation datasets — our toy dataset, the BraTS dataset, and the cryogenic electron microscopy dataset, demonstrates the robustness of our approach.

With the synthesis dataset, we tuned over a range of hyperparameter values using its validation set. This includes weight of triplet loss $\beta = \{10^{-2}, 10^{-1}, 1, 10^1, 10^2, 10^3, 10^4, 10^5\}$ with the final weight used as $\beta = 10^3$. The patch size for inference was tuned with range $p = \{5, 10, 15, 20, 40\}$ with chosen size as $5 \times 5 \times 5$. The number of epochs was tuned with range $e = \{3, 5, 8, 10, 12, 15\}$ with final epoch number $8$.

The BraTS 2019 dataset Menze et al. (2014); Bakas et al. (2017; 2018) can be downloaded following directions from `https://www.med.upenn.edu/cbica/brats2019/registration.html`. We will release our toy dataset with the final code release.

### A.9 MULTI-PATCH SAMPLING

Our method is designed to model the compositional hierarchy of 3D data, where we often find visual substructures contained within other structures. Based on this idea, we sample triplets of 3D volume patches that capture this notion of hierarchical structure. Triplets are sampled through the following process: First, we sample a 3D patch of data to be the anchor element, and consider this to be the parent in the triplet. Second, we sample a smaller patch of data that is completely contained within the parent patch, and consider this to be the positive child patch. Then, we sample a smaller patch of data that does not overlap with the anchor patch, and consider this to be the negative child patch. See Section 4.1 for further details on sampling procedure. We input the (parent, positive child, negative child) tuples into our hierarchical triplet loss, where the loss encourages the anchor parent and positive child to have closer representations relative to the anchor and the negative child. See Figure 5 for an overview.

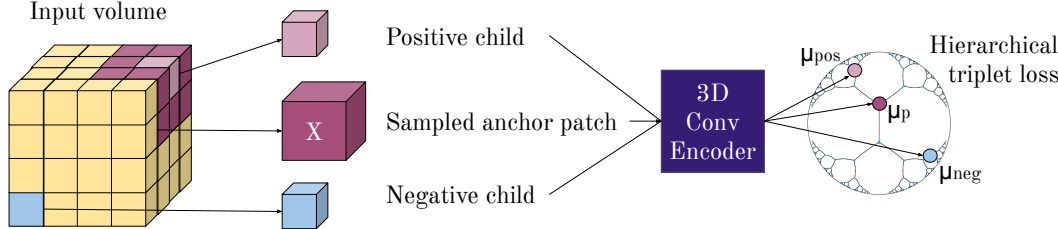

Figure 5: Example of multi-patch sampling procedure with sampled anchor patch, positive child, and negative child.

## A.10 LATENT DIMENSION ABLATION

In Section 5.1, Section 5.2, and Section 5.3, our experiments were all run with latent dimension of two. To show the effect of higher latent space dimensions, we report an ablation study for both hyperbolic and Euclidean representations. As expected, the performance increases with dimension for our model with Euclidean latent space, but our model with hyperbolic latent space still outperforms the Euclidean model at all tested dimensions.

Table 8: Ablation study of latent space dimension for Euclidean and Hyperbolic models on the toy dataset. Dice scores for all three levels are reported.

| Latent Space | Dice *Level* | d=2 | d=3 | d=5 | d=8 | d=16 |
|---|---|---|---|---|---|---|
| Euclidean | *Level 1* | 0.95211 | 0.95943 | 0.95574 | 0.94159 | 0.95350 |
| | *Level 2* | 0.54065 | 0.53827 | 0.54959 | 0.52889 | 0.54097 |
| | *Level 3* | 0.21623 | 0.21283 | 0.21850 | 0.22612 | 0.22791 |
| Hyperbolic | *Level 1* | 0.76111 | 0.83793 | 0.84664 | 0.87080 | 0.87210 |
| | *Level 2* | 0.34202 | 0.36218 | 0.37751 | 0.48133 | 0.49511 |
| | *Level 3* | 0.15349 | 0.17568 | 0.16543 | 0.22521 | 0.22767 |

