# OpenReview forum: "Learning Hyperbolic Representations for Unsupervised 3D Segmentation"
_ICLR.cc/2021/Conference — Reject_

### Official Review · AnonReviewer4 · 2020-10-28
**Learning Hyperbolic Representations for Unsupervised 3D Segmentation**

**Rating:** 3
**Confidence:** 3

**Review:**

The authors of this manuscript propose an unsupervised learning framework for 3D segmentation of biomedical images. Specifically, the proposed method learns effective representations for 3D patches using variational autoencoder (VAE) with a hyperbolic latent space. Its main contribution lies at that it introduces a new unsupervised learning framework including hyperbolic convolutional VAE and hierarchical triplet loss. This work conducts experiments on toy dataset, the Brain Tumor Segmentation dataset, and cryo-EM data. The experiment demonstrates competitive performance of the proposed method.

Major Strengths of The Paper
1. The idea of hyperbolic 3D convolutional VAE is interesting.
2. The experiment shows that this idea performs favorably against the state-of-the-art.

Major Weaknesses of The paper
1. The manuscript contains some grammatical errors, and the writing could also be improved. For example, there are some grammar mistakes like “our work is a first to introduce…”. The paper will be more clear if the authors could further polish the manuscript.
2. The proposed model is not novel enough. In my personal opinion, the proposed model generalizes the gyroplane layer and hyperbolic representations in previous literature to 3D biomedical images. The multi-scale sampling and triplet-loss are also commonly used mechanisms. It would be better if the authors could elaborate the novelty of the proposed model.
3. The biologically-inspired toy dataset does not look close to a biomedical dataset. Although we totally understand there should be differences between simulated dataset and real dataset, the toy dataset (i.e. composed of regular shapes) oversimplifies the biomedical image, and makes the experiment results less convincing. It will be better if the authors can simulate images close to real biomedical images, at least with irregular shapes or fuzzy boundaries.
4. For the baseline selection, the authors used the model in paper “Unsupervised Segmentation of Hyperspectral Images Using 3D Convolutional Autoencoders,” which is published in IEEE Geoscience and Remote Sensing Letters. There should be a domain gap between remote sensing images and biomedical images. I personally think it may not be a fair comparison for the baseline methods. It will be better if the authors could select some of the state-of-the-art unsupervised 3D segmentation methods as baselines.

---

> ### Author Response · Authors · 2020-11-19
> **Response to Reviewer 4**
>
> We thank the reviewer for pointing out that the idea of “hyperbolic 3D convolution VAE is interesting” and that our experiments “demonstrates competitive performance of the proposed method” .
>
> Re: Grammatical errors.
> We thank the reviewer for pointing this out and have edited the paper for both grammar and clarity.
>
> Re: Novelty.
> We would like to point out that our method tackles the challenging task of unsupervised 3D segmentation on complex 3D data, which requires additional representation discriminability compared to prior work tackling classification tasks on simple MNIST data. We demonstrate that our self-supervised hierarchical triplet loss and multi-patch sampling method on 3D voxel data shows significant improvement over our baselines, and that they are essential to capturing hierarchical structure in complex 3D images. In addition, our gyroplane convolutional layer allows for effective mapping of 3D image data from hyperbolic to Euclidean spaces. We have updated our introduction to better highlight our contributions, and thank the reviewer for the suggestion.
>
> Re: Toy dataset.
> We thank the reviewer for suggesting a more realistic biomedical dataset with irregular shapes. We would like to point out that BraTS, which we also evaluate on, is a real-world biomedical dataset that is a common benchmark for 3D image segmentation. We agree that a toy dataset with irregular shapes would also provide useful insight, but we note that it is non-trivial to generate these types of datasets such that they also satisfy our constraint of incorporating 3D hierarchy. However, we are working to create this dataset and hope to include this result in an updated version of the paper before the review period ends.
>
> Re: Unsupervised segmentation of hyperspectral images.
> We thank the reviewer for raising concerns on “Unsupervised Segmentation of Hyperspectral Images Using 3D Convolutional Autoencoders”. Even though this work was demonstrated for the geoscience field, the method is general and a reasonable approach for generic 3D voxel data, so we include it for comparison in addition to our other unsupervised baselines. To the best of our knowledge, we have provided a comprehensive evaluation against prior work in this field. As the field of unsupervised 3D segmentation for voxel data is still relatively unexplored, we have included comparisons to prior semi-supervised works as well for completeness. We hope this provides helpful context, and if there is an additional, specific method that the reviewer has in mind, we would appreciate a reference to it.

---

> > ### Author Response · Authors · 2020-11-23
> > **Update on irregular toy dataset**
> >
> > Re: Irregular dataset.
> > We thank the reviewer for suggesting the inclusion of this dataset. We have added an irregular dataset created through applying smooth noise to boundaries of each shape. Specifically, we generate smooth noise by first sampling random points in our voxel grid and random values according to a Gaussian distribution, and interpolate to yield smooth noise. We then use this smooth noise function to perturb the points that fall within the interior of the three largest shapes. Examples of the dataset can be seen in Figure 4.
> >
> > In Section A.5 of the Appendix, we show two additional tables. The first, Table 4,  is our comparison to prior work applied on the aforementioned irregular dataset, the second, Table 5, is an ablation study applied on the same irregular dataset, both with error bars over four independent runs, thanks to the suggestion of error bars by Reviewer 2.
> >
> > We note that in Table 4, our proposed method outperforms prior work significantly on the irregular dataset, following our initial observations from Table 1. We can see that while all methods show slight decrease in performance, our method is still able to maintain the lead in performance as compared to prior work across all levels.
> >
> > For ablations on our irregular toy dataset, results can be found in Table 5 in the paper. As with our original toy dataset, we find that our best models with hyperbolic latent space still outperform models with Euclidean latent space. We also demonstrate that the gyroplane convolutional layer and hierarchical triplet loss are both effective compared to the base hyperbolic configuration. However, despite it being effective compared to the base hyperbolic configuration, models with hyperbolic hierarchical triplet loss performed less well across the board as compared to the original toy dataset. We hypothesize that this is due to the specific challenges that the irregular dataset brings, for example, needing to recognize noisy instances of irregular shape as the same class in Levels 2 and 3. Therefore, our proposed gyroplane convolutional layer by itself is able to add more effective learning capacity, and shows significant improvement. The added hierarchical triplet loss performs less well in the irregular dataset than in our original toy dataset because in our multi-patch sampling method, each patch is sampled at random capturing parts of the 3D input. Since the boundary of the shape changes in every image, with random sampling learning is more difficult for our hierarchical triplet loss. We don’t see the same phenomenon for Level 1 since background/foreground segmentation is an easier task. We conclude that with the level of irregularity added to our dataset (examples shown in Figure 4), the gyroplane convolutional layer with the hyperbolic latent space provides more effectiveness than the triplet loss.
> >
> > We also note that in real-world datasets, such as in our work in cryogenic electron microscopy, the overall shapes of each class of object is similar, and do not contain such dramatic irregularity. For example, vesicles are almost-circular ellipses with only slight eccentricity (deformations with slight stretch), but without distinctive irregularities and protrusions in our irregular dataset. Overall, our experiments demonstrate that different components of our method are useful for different scenarios, and that our method overall robustly outperforms prior work across data with different characteristics. All hyperbolic configurations of our method seen in Table 4 outperform past unsupervised methods, and our approach of learning hyperbolic representations of complex 3D data for segmentation is more effective than methods with canonical Euclidean representations.

---

### Official Review · AnonReviewer1 · 2020-10-29
**Interesting paper but needs clarifications for experiments**

**Rating:** 7
**Confidence:** 3

**Review:**

The paper considers learning hyperbolic representations for unsupervised 3D segmentation. Since the general task of producing annotations for 3D data can be expensive (e.g. for segmentation in dense voxel grids), this is an important problem. The paper proposes to learn hierarchical data structures (e.g. 3D biomedical images) with a hyperbolic variational autoencoder. The paper adapts different metric learning approaches, such as triplet loss and computing a Frechet mean on Riemannian manifolds for clustering.

The paper is easy to read and interesting to the computer vision and biomedical imaging communities. In terms of machine learning theory, there is not much novelty since the paper combines different existing approaches into a single framework. However, the paper tackles a difficult problem which is the representation of hierarchical 3D data with a simple solution.

- I think that the self-supervised aspect of the approach in the hierarchical triplet loss paragraph is important. However, there does not seem to be much emphasis on it, and I did not fully understand how the positive and negative child patches are generated. Could you please clarify that part and maybe illustrate it in a figure?

- Since the approach is used in the unsupervised setup, how are hyperparameters tuned? There is a discussion about hyperparameter tuning in Section A.7 but the setup is not clear to me. Do you use annotations/labels for the validation set? What is the stopping criterion? How do you pick the best set of hyperparameters? This needs clarification.

- It is nice to see an ablation study of different components of the Euclidean vs hyperbolic approach in Table 2. However, it is known that hyperbolic representations perform better than Euclidean representations to representation trees in low-dimensional space. In high-dimensional space, Euclidean and hyperbolic representations generally perform similarly.
If I understand correctly, the dimensionality of the wrapped normal distribution is 2 for the toy dataset. Could we have an ablation study of the impact of higher dimensions for both hyperbolic and Euclidean representations? This would be particularly interesting for real-world applications.
What is the dimensionality of the latent space in Section 4.2 and 4.3?

In conclusion, I think that the paper is interesting, but needs some clarifications.

Minor comment: representation dimensionality d and distance d use the same font.

---

> ### Author Response · Authors · 2020-11-19
> **Response to Reviewer 1**
>
> We thank the reviewer for pointing out that our paper addresses an “important problem” and that the paper is “easy to read and interesting to the computer vision and biomedical imaging communities”.
>
> Re: Multi-patch sampling.
> We apologize for any confusion arising from our current description of self-supervised training. Our method is designed to model the compositional hierarchy of 3D data, where we often find visual substructures contained within other structures. Based on this idea, we sample triplets of 3D volume patches that capture this notion of hierarchical structure. Triplets are sampled through the following process: first, we sample a 3D patch of data to be the anchor element, and consider this to be the parent in the triplet. Second, we sample a smaller patch of data that is completely contained within the parent patch, and consider this to be the positive child patch. Then, we sample another smaller patch of data that does not overlap with the anchor patch, and consider this to be the negative child patch. See Section 4.1 for further details on sampling sizes. We input the (parent, positive child, negative child) tuples into our hierarchical triplet loss, where the loss encourages the anchor parent and positive child to have closer representations relative to the anchor and the negative child. We thank the reviewer for pointing this out and have included additional explanation as well as an additional figure in Section A.9 of the Appendix.
>
> Re: Hyperparameters.
> We also thank the reviewer for suggesting more clarifications on hyperparameter tuning. We’ve updated our hyperparameter tuning explanation in Section A.8 of the Appendix. We use a single set of hyperparameters on all of our evaluation datasets, and these hyperparameters are not tuned on any of the evaluation datasets. In order to obtain a reasonable set of hyperparameters, we created a completely separate synthetic dataset on which we trained models and tuned hyperparameters. This synthetic dataset was created in a similar manner to our toy dataset; however, we designed it to have different and fewer objects, simpler nesting structure, no noise, and fewer textures. The application of this single set of hyperparameters to our evaluation datasets --- our toy dataset, the BraTS dataset, and the cryogenic electron microscopy dataset, demonstrates the robustness of our approach.
>
> Re. Latent dimension experiments.
> We thank the reviewer for the suggestion. We’ve added an ablation study of latent space dimension in Section A.10 of the Appendix. Our experiments were all run with latent dimension of 2 for Sections 5.1, 5.2, and 5.3. To show the effect of higher latent space dimensions, we report an ablation study for both hyperbolic and Euclidean representations.  As expected, the performance increases with dimension for our model with Euclidean latent space, but our model with hyperbolic latent space still outperforms the Euclidean model at all tested dimensions.
>
> Re. Minor comment on font.
> We appreciate the comment and have addressed it.

---

> > ### Comment · AnonReviewer1 · 2020-11-24
> > **Thank you for the clarifications**
> >
> > Thank you for the clarifications. I will keep my scores although I agree with other reviewers that there is not much novelty in terms of theory since the model is an aggregation of different existing methods into a single pipeline.
> > I am not an expert in biomedical imaging so I can't judge the relevance of the approach for biomedical imaging.

---

### Official Review · AnonReviewer2 · 2020-10-30
**An interesting result with a high practical relevance to neuroimaging**

**Rating:** 7
**Confidence:** 5

**Review:**

The paper proposes to train a variational autoencoder with a hyperbolic hidden layer augmenting the ELBO loss with a hierarchical triplet loss between nested subsets of data suitable for spatial data structured hierarchically. This process results in representations that can be clustered and used for unsupervised segmentation.

# Strengths

Obtaining segmentation labels for 3D data is notoriously difficult and error-prone as even the experts are unable to stay focused for a very long time required to label voxels in a large MRI volume, as an example. Practical advances in unsupervised labeling are a welcome addition to application areas relying on 3D segmentation.

This paper is able to exploit the recently developed approach to hyperbolic convolutions in a variational autoencoder training with a simple addition of a hierarchical triplet loss. Notably, this simple idea leads to performance improvements over other existing unsupervised or self-supervised methods.

Demonstrations on synthetic data and a couple of real datasets, together with comparison to existing methods is a plus that helps to understand the value of the proposed approach.

# Weaknesses and comments

The limited technical novelty of the contribution, as most of the innovations are although recent but already published elsewhere.

The application component is not as rigorously evaluated as one would need to assess the real practical applicability of the approach. For example, in the case of the tumor, the size may bear a large meaning, and volume error in addition to the DICE coefficient would provide much-needed information. For electron microscopy, tomograms or even for the tumor to be treated with radiation therapy the Hausdorff distance may be an informative and potentially more appropriate metric of comparison. Basically, it is very difficult to assess experiments in any meaningful way besides - "this might be interesting".

Neither of the measurements provides error bars (all tables), so it is impossible to assess the uncertainty (variability of the approach) and for values that are not too far apart estimate if the prediction is even statistically different.

It would be good to discuss the training process, hyperparameter search, and computational complexity of the proposed method.

Could the encoder be trained directly using the triplet loss without invoking the VA at all?

---

> ### Author Response · Authors · 2020-11-19
> **Response to Reviewer 2**
>
> We thank the reviewer for pointing out “practical advances … are a welcome addition to application areas relying on 3D segmentation” as well as “demonstrations on synthetic data and … real datasets helps to understand the value of the proposed approach”.
>
> Re: Technical contribution.
> We would like to first address the technical novelty of the contribution. We note that we are the first to leverage hyperbolic representations for complex 3D data as opposed to simple graphs or MNIST data, as well as for the difficult tasks of 3D segmentation as opposed to classification. These new challenges require innovative methods to learn more representation discriminability, such as our hierarchical triplet loss, multi-scale sampling scheme as well as our gyroplane convolutional layer. We show the success of these methods in our ablation studies. We have edited the introduction to further clarify these contributions.
>
> Re: Hausdorff distance.
> We appreciate the reviewer's suggestion. We have added average and 95th percentile Hausdorff distance between the predicted tumor and ground truth tumor as a complementary evaluation metric on the BraTS dataset in Table 7 Section A.6 of the Appendix. We compare our model against all of our unsupervised baselines, and find that our model outperforms all other methods on both metrics.
>
> Re: Error bars.
> We also thank the reviewer for suggesting error bars for assessing variability of our approach. We have added error bars for both the comparison to prior work as well as the ablation study on the irregular toy dataset suggested by Reviewer 4 in Table 4 and Table 5 of Section A.5 in the Appendix. In addition, we have also added error bars to all of our model variants in the BraTS ablation study suggested by Reviewer 3 in Table 6 of Section A.6 in the Appendix. The error bars are generated over 4 independent runs.
>
> Re: Training process.
> We discuss the training process and hyperparameter search in Section A.8 of the Appendix, and computational complexity in Section A.5 of the Appendix. We apologize for the confusion and have added more details to both sections to clarify.
>
> Re: Triplet loss only.
> We appreciate the reviewer’s suggestion. In early stages of our research, we did explore just using a simple encoder in hyperbolic space without the VAE, however, we found that this yields lower results and did not perform as well, therefore we have included the VAE as our base model and built our entire work on this premise.

---

### Official Review · AnonReviewer3 · 2020-11-03
**An interesting research topic in the medical field but limited clinical use in practice**

**Rating:** 4
**Confidence:** 3

**Review:**

Overall, the unsupervised 3D segmentation task tackles one major pain point of the medical imaging field, where the professional annotation is hard and expensive to obtain. This work adopts 3D VAE to map 3D patches into a hyperbolic latent space, then apply clustering followed by the Hungarian algorithm to finish the segmentation task. Though with merits on the introduced hyperbolic space and the gyroplane conv, I have the following concerns and questions.

Although I totally appreciate the exploration of the fully unsupervised method to do the 3D segmentation to tackle the lacking of annotation problem, my major concern is that a reliable medical imaging system cannot be obtained through this way and won't be clinically meaningful. Results in Table 1 and Table 3 shows that the 3D semi-supervised by training a vanilla 3D UNet only on 2% of the ground-truth data can perform much better than the proposed unsupervised method. I believe even this simple semi-supervised way is closer to the practical scenario, not to mention that a medical AI system needs to be as reliable as possible for doctors to use.

My questions:
1) What is the running time comparison between the between the Hyperbolic space and the Euclidean space?
2) Can the ablation studies done on the BraTS data? I would personally think this way can be more serious and convincing rather than on a synthesized toy dataset.

---

> ### Author Response · Authors · 2020-11-19
> **Response to Reviewer 3**
>
> We thank the reviewer for pointing out that our method “tackles one major pain point of the medical imaging field, where professional annotation is hard and expensive to obtain”, and introduces “merits on the introduced hyperbolic space and the gyroplane conv”.
>
> Re: Concerns that a reliable medical imaging system cannot be obtained.
> We appreciate the discussion on clinically meaningful usage, and that semi-supervised methods using some amount of labelled data can outperform completely unsupervised methods. However we would like to emphasize that there are many scenarios where unsupervised methods are still desirable. First, annotating even 2% of data is time-consuming especially as the number of desired types of structure to segment increases. In some applications, a tradeoff to have comparable or slightly lower accuracy, but segmentations of more types of structures, is better. In our own research, we are using these types of methods not to assist clinicians with medical diagnosis, but to study differences in anatomical structure across large patient populations to develop targeted treatment for disease based on patient anatomical characteristics. Second, in many biomedical applications such as research in cryogenic electron microscopy (cryo-EM), annotations are also impossible to obtain due to our lack of knowledge of the complex data. Therefore, our goal is to create a method capable of unbiased discovery of novel cellular structures that repeat across from 3D voxel images, as in some cases, human annotations may introduce undesirable bias into segmentations. Some of our own research focused on cryo-EM directly has this need; in Figure 3 of our paper, we show an example of discovery in cryo-EM. We have also revised our introduction to clarify this; thank you for the feedback.
>
> Re: Running time comparison.
> We have added runtime comparisons between hyperbolic and Euclidean models in Section A.5 of the Appendix. Our implementations of the proposed models take between five to eight hours to train on a single Titan RTX GPU for both Euclidean and Hyperbolic variants. We note that for our current implementation, hyperbolic k-means clustering takes on the order of a few hours versus minutes for Euclidean k-means. However, this is because we are using our own unoptimized implementation based on recent research in Frechet mean algorithms, and standard packages such as scikit-learn do not include hyperbolic k-means algorithms. The Euclidean k-means algorithms in these packages are heavily optimized with parallelization. We anticipate that such optimization would bring the hyperbolic k-means’s runtime to the order of the Euclidean k-means, as the computational complexity of the algorithms are similar in practice.
>
> Re: Ablation study.
> We thank the reviewer for suggesting ablations studies on the BraTS dataset. We have added this ablation study in Section A.6 of the Appendix. In addition, we added error bars over 4 independent runs to measure performance variability; we thank Reviewer 2 for the suggestion of error bars. We follow the same ablation study from Section 5.1. In Table 6 of Section A.6 of the Appendix, we can see that our best Hyperbolic model outperforms our best Euclidean model significantly. The addition of the triplet loss improves both Euclidean and Hyperbolic models, while Hyperbolic models see more improvement due to their ability to better encode hierarchy. Our gyroplane convolutional layer also improves performance, while both of our additions jointly improve upon our Hyperbolic baseline, showing the benefit of these added components to learning effective representations.

---

### Decision · Program_Chairs · 2021-01-07
**Final Decision**

**Decision:**

Reject

**Comment:**

This application paper applies hyperbolic convolutions in VAE learning to perform unsupervised 3D segmentation.
Addition of these components enables performance improvements in the unsupervised segmentation task.
Overall, the paper is borderline and the reviewers mention the limited novelty of the approach, which largely uses components that have been developed before. Even though the paper presents an application of these methods to a relevant and significantly more challenging task than prior work, I recommend rejection from ICLR due to concerns about novelty.